# A nuclear-based quality control pathway for non-imported mitochondrial proteins

**Viplendra PS Shakya, William A Barbeau, Tianyao Xiao, Christina S Knutson, Max H Schuler, Adam L Hughes***

Department of Biochemistry, University of Utah School of Medicine, Salt Lake City, United States

**Abstract** Mitochondrial import deficiency causes cellular toxicity due to the accumulation of non-imported mitochondrial precursor proteins, termed mitoprotein-induced stress. Despite the burden mis-localized mitochondrial precursors place on cells, our understanding of the systems that dispose of these proteins is incomplete. Here, we cataloged the location and steady-state abundance of mitochondrial precursor proteins during mitochondrial impairment in *Saccharomyces cerevisiae*. We found that a number of non-imported mitochondrial proteins localize to the nucleus, where they are subjected to proteasome-dependent degradation through a process we term nuclear-associated mitoprotein degradation (mitoNUC). Recognition and destruction of mitochondrial precursors by the mitoNUC pathway requires the presence of an N-terminal mitochondrial targeting sequence and is mediated by combined action of the E3 ubiquitin ligases San1, Ubr1, and Doa10. Impaired breakdown of precursors leads to alternative sequestration in nuclear-associated foci. These results identify the nucleus as an important destination for the disposal of non-imported mitochondrial precursors.

## Introduction

Mitochondrial dysfunction is a hallmark of aging and associated with many age-related and metabolic diseases (*Wallace, 2005*). Mitochondrial impairment disrupts metabolic pathways housed within the mitochondrion and also prevents the import of thousands of mitochondrial resident proteins that rely on an efficient mitochondrial membrane potential for translocation into the organelle (*Martin et al., 1991*; *Pagliarini et al., 2008*; *Wiedemann and Pfanner, 2017*). Recent studies have shown that non-imported mitochondrial precursor proteins are toxic for cells and identified several cellular pathways that combat this stress (*Boos et al., 2019*; *Boos et al., 2020*; *Hansen et al., 2018*; *Itakura et al., 2016*; *Mårtensson et al., 2019*; *Wang and Chen, 2015*; *Weidberg and Amon, 2018*; *Wrobel et al., 2015*). These systems, which are collectively referred to as the mitoprotein-induced stress response (*Boos et al., 2020*), protect cells during times of mitochondrial import deficiency through a variety of mechanisms, including downregulation of cellular translation (*Wang and Chen, 2015*); upregulation of proteasome and chaperone capacity through a process termed UPRam (unfolded protein response activated by mistargeting of proteins) (*Boos et al., 2019*; *Wrobel et al., 2015*); destruction of non-imported mitochondrial precursors at the mitochondrial surface through mitochondrial-associated degradation (MAD) (*Metzger et al., 2020*), mitochondrial translocation-associated degradation (mitoTAD) (*Mårtensson et al., 2019*), mitochondrial-compromised import response (mitoCPR) (*Weidberg and Amon, 2018*), and mitochondrial-associated ribosome quality control (mitoRQC) (*Izawa et al., 2017*; *Su et al., 2019*; *Zurita Rendón et al., 2018*); ubiquilin-dependent triaging of mitochondrial membrane proteins for degradation (*Itakura et al., 2016*); and endoplasmic sequestration followed by either destruction via the conserved protein Ema19 (*Laborenz et al., 2021*) or re-delivery of non-imported mitochondrial precursors via an endoplasmic reticulum (ER) surface retrieval pathway (ER-SURF) (*Hansen et al., 2018*).

**\*For correspondence:**
hughes@biochem.utah.edu

**Competing interests:** The authors declare that no competing interests exist.

Despite the recent advances in this field, only a fraction of the non-imported mitochondrial proteome has been analyzed under conditions of mitochondrial impairment. Thus, our understanding of the fate of non-imported mitochondrial precursors and the systems that regulate their disposal is likely incomplete. Here, using microscopy and immunoblot-based screens in *Saccharomyces cerevisiae*, we find that non-imported mitochondrial proteins accumulate in many regions of the cell upon mitochondrial depolarization and identify the nucleus as a previously unrecognized quality control destination for non-imported mitochondrial precursors. We show that many mitochondrial proteins localize to the nucleus upon import failure, where they are subjected to proteasome-dependent destruction via redundant action of the E3 ubiquitin ligases San1, Ubr1, and Doa10. When degradation capacity is exceeded, mitochondrial precursors are sequestered into nuclear-associated protein foci.

Interestingly, we determined that the N-terminal mitochondrial targeting sequence (MTS) (*Omura, 1998*) is necessary for non-imported precursor protein induced-toxicity, degradation, and sequestration into nuclear-associated foci, but dispensable for nuclear localization. The presence of an MTS is also required for degradation and toxicity of non-imported proteins that localize to cellular regions other than the nucleus, implicating the MTS as a major driver of non-imported precursor toxicity. Finally, we show that nuclear accumulation of non-imported precursors arises during cellular aging. Overall, this work demonstrates that non-imported mitochondrial proteins exhibit numerous fates within cells and outlines a nuclear-based quality control pathway for non-imported mitoproteins we refer to as mitoNUC (nuclear-associated mitoprotein degradation).

## Results

### The non-imported version of the mitochondrial protein Ilv2 localizes to the nucleus

We previously showed that the mitochondrial network undergoes extensive fragmentation and depolarization during replicative aging in budding yeast, which is defined as the number of times an individual yeast cell undergoes division (*Hughes and Gottschling, 2012*). In our prior work, we utilized an endogenously tagged version of the mitochondrial outer membrane (OM) protein Tom70-mCherry to visualize the mitochondrial network. In contrast to Tom70, which does not rely on mitochondrial membrane potential for its mitochondrial localization (*Söllner et al., 1990*), functional, endogenously GFP-tagged Ilv2 (*Figure 1—figure supplement 1A*), a mitochondrial matrix enzyme in isoleucine and valine biosynthesis (*Falco et al., 1985*), exhibited dual localization in replicatively aged yeast cells. In addition to a pattern consistent with mitochondrial tubules, Ilv2-GFP also localized to the nucleus in over 80% of aged cells, as indicated by diffuse GFP fluorescence within a region surrounded by the nuclear pore protein Nup49-mCherry (*Figure 1A*).

Mitochondrial depolarization is a hallmark feature of aged yeast (*Hughes and Gottschling, 2012*). Because depolarization is prominent, and the import of Ilv2 requires a mitochondrial inner membrane (IM) potential, we wondered whether the fraction of Ilv2-GFP localized to the nucleus represented a non-imported precursor pool of this protein. Consistent with that idea, treatment of cells with the mitochondrial IM depolarizing agent trifluoromethoxy carbonyl cyanide phenylhydrazone (FCCP) (*Parker, 1965*) also caused Ilv2-GFP accumulation in the nucleus, which was marked with 4′,6-Diamidine-2′-phenylindole dihydrochloride (DAPI) (*Figure 1B*). Nuclear localization did not result from GFP-tagging, as indirect immunofluorescence showed a similar nuclear localization for native Ilv2 (*Figure 1C*) as well as functional C-terminally FLAG-tagged Ilv2 (*Figure 1—figure supplement 1A*) in FCCP-treated cells (*Figure 1—figure supplement 1B*). Furthermore, Ilv2-GFP localized to the nucleus in cells conditionally depleted of the essential OM protein import channel Tom40 (*Mnaimneh et al., 2004*; *Vestweber et al., 1989*; *Figure 1—figure supplement 1C,D*), indicating that nuclear localization was not caused by off-target effects of FCCP, but was specific to defects in mitochondrial protein import.

We hypothesized that the nuclear pool of Ilv2 likely represented a fraction of the protein that could not be imported into mitochondria. Consistent with that idea, western blot analysis revealed that higher-molecular-weight forms of Ilv2, Ilv2-GFP, and Ilv2-HA accumulated in cells treated with FCCP (*Figure 1D,E*, *Figure 1—figure supplement 1E*). Mitochondrial proteins such as Ilv2 are synthesized with an N-terminal MTS extension that is proteolytically removed from the mature peptide

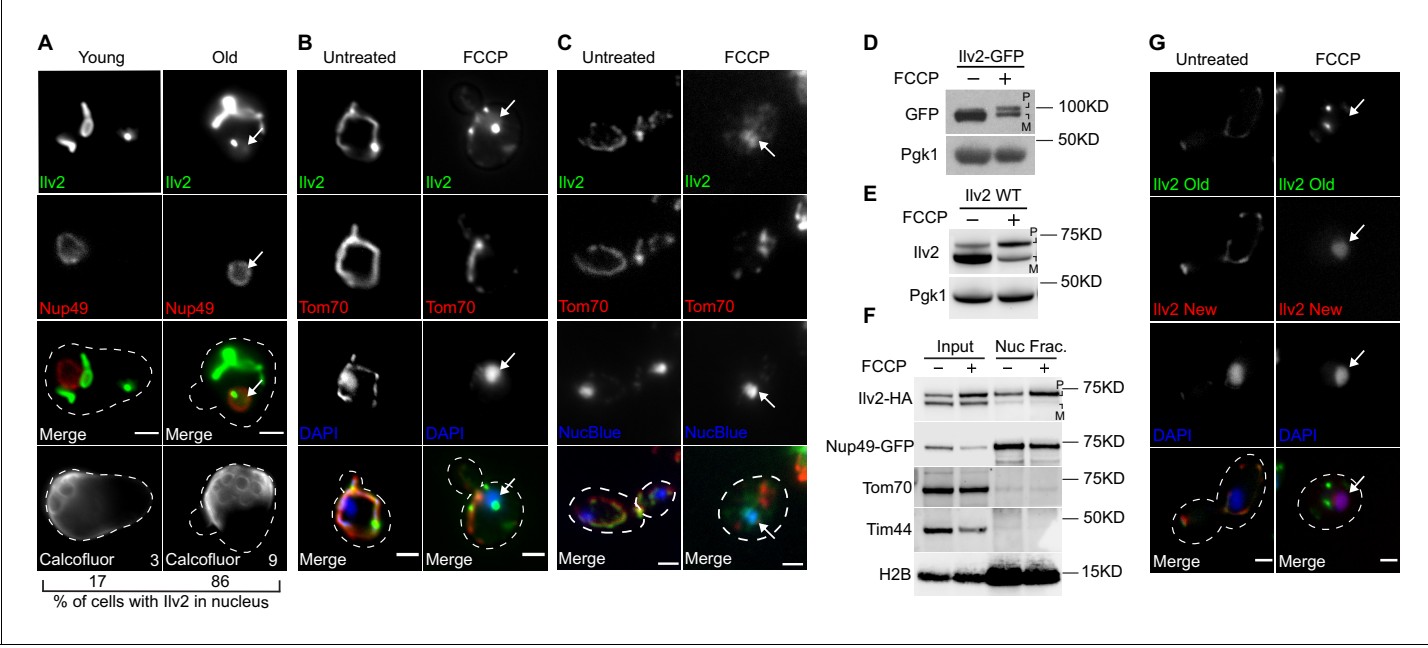

**Figure 1.** Non-imported mitochondrial protein Ilv2 localizes to the nucleus upon import failure. (**A**) Representative images of old and young yeast expressing the indicated Ilv2-GFP and nuclear marker Nup49-mCherry. Percentage of cells with Ilv2 in the nucleus (n = 30 cells) and age of representative cell (determined by bud scar counting) are indicated in bottom panels. Bud scars stained with calcofluor. (**B**) Yeast expressing Ilv2-GFP and Tom70-mCherry ± FCCP. (**C**) Indirect immunofluorescence for endogenous Ilv2 of yeast expressing Tom70-mCherry. (**D**) Western blots of yeast expressing Ilv2-GFP ± FCCP. (**E**) Western blots of endogenous Ilv2 in yeast treated ± FCCP. Pgk1 = loading control in all instances. (**F**) Western blot showing enrichment of the precursor form of Ilv2-HA in the nuclear fraction. Nup49-GFP and H2B = nuclear markers, Tom70 and Tim44 = mitochondrial markers. (**G**) RITE-tagged cells treated with β-estradiol at time of FCCP addition to initiate Cre/lox switching of Ilv2 epitope tag from GFP (old) to RFP (new). (**B**, **C**, and **G**) Nucleus stained with DAPI. P = precursor, M = mature. All scale bars = 2 μm. Arrows denote nucleus.

The online version of this article includes the following figure supplement(s) for figure 1:

**Figure supplement 1.** Mitochondrial protein Ilv2 localizes to the nucleus upon import failure.

only after they transit the mitochondrial IM (*Mossmann et al., 2012*; *Pfanner et al., 2019*). Thus, the higher-molecular-weight form of Ilv2 in FCCP-treated cells likely represents the immature, precursor form of the protein. In support of the idea that the non-imported pool of Ilv2 localizes to the nucleus, the precursor form of Ilv2-HA was specifically enriched in nuclear fractions isolated from FCCP-treated cells, while other mitochondrial proteins, including Tom70, Tim44, as well as the mature form of Ilv2-HA, were excluded (*Figure 1F*). Additionally, we utilized the Recombination-Induced Tag Exchange (RITE) system (*Verzijlbergen et al., 2010*), which consists of an estradiol-induced Cre recombinase in combination with gene-targeted loxP sites to facilitate on-demand epitope-tag switching upon exposure of cells to estradiol, to examine the fate of both old and newly synthesized Ilv2 in the same cell. We found that only newly synthesized Ilv2 (Ilv2 New, RFP tagged) localized to the nucleus upon FCCP treatment, while Ilv2 already present in mitochondria (Ilv2 Old, GFP tagged) did not (*Figure 1G*). Collectively, these results indicate that when the translocation of Ilv2 into mitochondria is blocked by genetic or pharmacologic impairment of mitochondrial import, the non-imported precursor form of Ilv2 alternatively localizes to the nucleus.

## The nucleus in one of several fates for non-imported mitochondrial proteins

We next sought to determine the extent to which non-imported proteins localize to the nucleus in cells lacking efficient mitochondrial import. To address this question in a systematic manner, we imaged a collection of yeast strains expressing 523 distinct mitochondrial proteins with carboxy-terminal GFP fusions from their endogenous loci in the absence or presence of FCCP. These strains were derived from the yeast GFP collection (*Huh et al., 2003*) and co-expressed Tom70-mCherry, a mitochondrial OM marker that localizes to mitochondria independently of the membrane potential

(*Hughes et al., 2016*; *Söllner et al., 1990*). We found that 6.4% of the mitochondrial proteins analyzed behaved like Ilv2, exhibiting nuclear localization in FCCP-treated cells (class 1, *Figure 2A* and *Supplementary file 1*). Additionally, we identified four other major outcomes for mitochondrial proteins after membrane depolarization (*Figure 2A* and *Supplementary file 1*). These included continued localization to the mitochondrion (class 2, e.g., Tom20, 8.2% of all proteins), accumulation in the cytoplasm (class 3, e.g., Acp1, 36.1% of all proteins), localization to the ER (class 4, e.g., Mir1, 2.7% of all proteins), and reduced overall abundance to the point of being nearly undetectable (class 5, e.g., Cox15, 42.2% of all proteins). A subset of proteins (4.4%) localized to regions of the cell distinct from these five major classes upon FCCP treatment and associated with unidentified cellular

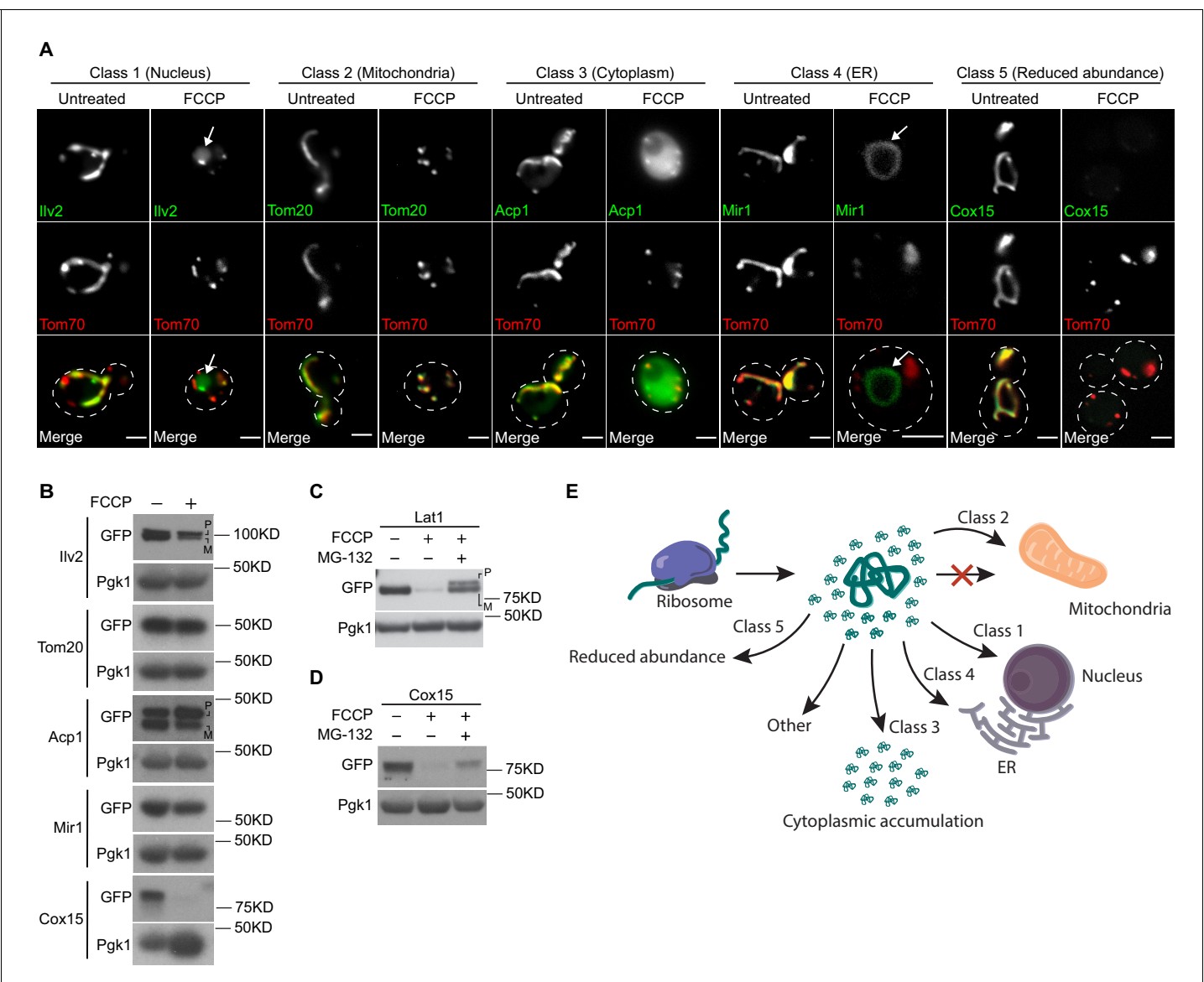

**Figure 2.** The nucleus in one of several fates for non-imported mitochondrial proteins. (**A**) Yeast expressing the indicated mCherry or GFP-tagged mitochondrial proteins ± FCCP. (**B–D**) Western blots of yeast expressing the indicated GFP-tagged mitochondrial proteins ± FCCP or ± FCCP ± MG-132. (**E**) Summary of non-imported mitochondrial protein fates. All scale bars = 2 µm. Arrows denote nucleus (**A**, class 1) or ER (**A**, class 4). P = precursor and M = mature.

The online version of this article includes the following figure supplement(s) for figure 2:

**Figure supplement 1.** Non-imported mitochondrial proteins have several distinct fates.
**Figure supplement 2.** Non-imported mitochondrial proteins have numerous fates.

membranes and foci (*Supplementary file 1*). In the majority of cases across protein classes, a portion of each protein localized to the mitochondrial network in addition to its new location, likely stemming from either residual mature form still present in the mitochondrion or continued delivery of a fraction of the precursor pool to the mitochondrial network (see *Figure 2A* images).

We validated representatives from our screen in several ways. Frist, we confirmed the ER localization of class 4 proteins via co-localization with the ER marker Sec61-mCherry (*Figure 2A*, *Figure 2— figure supplement 1A*). We also demonstrated via indirect immunofluorescence that untagged, native protein representatives from class 1 (nuclear localization), class 2 (continued localization to the mitochondrion) and class 4 (ER localization) exhibited identical localizations as their GFP-tagged counterparts (*Figure 2—figure supplement 1B*). Furthermore, we showed that FLAG-tagged versions of representatives from all classes of proteins also behaved identically to their GFP counterparts (*Figure 2—figure supplement 1C*) and determined that conditional depletion of the essential OM protein import channel Tom40 led to similar protein fates as treatment of cells with FCCP (*Mnaimneh et al., 2004*; *Vestweber et al., 1989*; *Figure 2—figure supplement 1D*). While we have not been able to validate our entire screen list using native antibodies, our results suggest that the observed changes in mitochondrial protein localization for representatives from each class are not caused by the presence of a GFP tag or off-target effects of FCCP.

We concurrently analyzed steady-state protein abundance via western blotting of the same set of GFP-tagged mitochondrial proteins in the absence and presence of FCCP, as this approach provided useful information about the state of Ilv2 in the nucleus. In general, steady-state levels of proteins localized to the mitochondrion, cytoplasm, and ER were unchanged or partially reduced with FCCP (*Figure 2B*, *Supplementary file 1*).

Proteins that localized to the nucleus or became undetectable often either moderately or strongly decreased in abundance upon FCCP treatment, respectively (*Figure 2B*, *Supplementary file 1*). We confirmed the changes in steady-state abundance for representatives from each class using antibodies against native forms of the proteins, as well as in strains expressing C-terminally HA-tagged versions of the proteins. We observed similar steady-state abundance changes as compared to our results for each GFP-tagged protein we retested (*Figure 2—figure supplement 2A–C*). Interestingly, further analysis indicated that the decline in class 5 protein abundance was either completely or partially blunted by proteasome inhibition via MG-132 depending on the individual protein substrate (*Figure 2C,D*, *Figure 2—figure supplement 2D,E*), implicating the proteasome in their destruction. Also, as with Ilv2, precursor forms of Acp1 and Lat1 (class 3 and 5) were visible in the presence of FCCP (*Figure 2B,C*).

Overall, our screen revealed several patterns amongst the proteins that comprised each screen class, and many of our observations aligned well with those from previous studies (*Figure 2E*). Nuclear-localized class 1 proteins were predominantly mitochondrial matrix enzymes, including numerous members of the TCA cycle. Most class 2 proteins that continued to localize to depolarized mitochondria were mitochondrial OM proteins that do not require a membrane potential for mitochondrial targeting (*Wiedemann and Pfanner, 2017*). Class 3 (cytoplasm) proteins were largely soluble proteins, several of which (e.g., Idh1, Idh2, Mss116, and Cis1) were previously found to be enriched in cytosolic extracts isolated from mitochondrial import-deficient yeast (*Wang and Chen, 2015*; *Weidberg and Amon, 2018*). ER-localized class 4 proteins were generally integral IM and OM proteins, some of which were previously reported to localize to the ER in cells with compromised mitochondrial import (*Hansen et al., 2018*). Finally, class 5, the largest of the classes, consisted of both soluble and membrane-bound mitochondrial proteins. These results indicate that numerous fates exist for non-imported mitochondrial proteins and that the nucleus represents a previously unrecognized destination for mitochondrial precursors during import failure.

## Nuclear protein quality control clears non-imported mitochondrial proteins

We next sought to understand the basis for the nuclear localization of non-imported mitochondrial proteins in the absence of functional mitochondrial import. The eukaryotic nucleus harbors a large proportion of cellular proteasomes and is a quality control destination for misfolded proteins (*Enam et al., 2018*). Because the overall abundance of nuclear-localized mitochondrial proteins declined during FCCP treatment, we tested whether non-imported mitochondrial precursor proteins were directed to the nucleus for proteasomal degradation. In support of that idea, the decline in

steady-state levels of Ilv2-GFP and Ilv2-HA upon FCCP treatment was blunted in the presence of proteasome inhibitor MG-132 (*Figure 3A*, *Figure 3—figure supplement 1A*). Ilv2 decline was also prevented in strains lacking a combination of three E3 ubiquitin ligases that operate in nuclear-associated protein quality control, San1 (*Gardner et al., 2005*; *Samant et al., 2018*), Ubr1 (*Prasad et al., 2010*), and Doa10 (*Deng and Hochstrasser, 2006*; *Swanson et al., 2001*) (E3 KO) (*Figure 3B*, *Figure 3—figure supplement 1B,C*). This result was true for GFP-tagged, HA-tagged, and untagged endogenous Ilv2. No combination of single or double knockouts completely prevented loss of Ilv2 upon mitochondrial depolarization, suggesting these ligases act redundantly to promote non-imported mitochondrial protein clearance (*Figure 3—figure supplement 1D*). Importantly, the addition of proteasome inhibitor or deletion of the aforementioned E3 ligases each led to a marked elevation in the higher-molecular-weight precursor form of Ilv2 in the presence of FCCP, suggesting the immature, Ilv2 precursor was the form of the protein specifically marked for proteasome clearance (*Figure 3A,B*, *Figure 3—figure supplement 1A–C*). In line with this observation, cycloheximide-chase analysis demonstrated that the half-life of the Ilv2, Ilv2-HA, and Ilv2-GFP precursor forms were altered in the E3 KO strain, while the mature forms were unaffected (*Figure 3C*, *Figure 3—figure supplement 1E,F*). Furthermore, ubiquitin immunoprecipitation assays indicated that Ilv2 was ubiquitylated in the presence of FCCP in a San1-, Ubr1-, and Doa10-dependent manner (*Figure 3D*). Proteasome-dependent degradation of a non-nuclear class 5 substrate (Lat1) was unaffected in the E3 KO strain, indicating that additional E3 ligases promote clearance of non-nuclear localized mitochondrial precursors (*Figure 3—figure supplement 1G*). Moreover, we found that our observations extend beyond Ilv2, as GFP-tagged versions of two other nuclear candidates identified in our screen, Dld1 and Dld2, (*Figure 3—figure supplement 2A–C*) were also eliminated in a proteasome and San1/Ubr1/Doa10-dependent manner (*Figure 3—figure supplement 2A–F*). Thus, a subset of non-imported mitochondrial proteins are subjected to nuclear-associated protein quality control when their import into mitochondria is impaired.

## Non-imported mitochondrial precursors are toxic and localize to nuclear-associated foci when clearance is impaired

As the toxicity of non-imported precursor proteins is well documented (*Wang and Chen, 2015*), we wondered whether failure to destroy nuclear-localized non-imported precursors would compromise cellular health. To test this idea, we compared the growth of wild type and the aforementioned E3 KO strains in the absence and presence of FCCP. We observed no growth defect in single, double, or triple E3 ligase knockout strains (*Figure 4A*, *Figure 4—figure supplement 1A*), suggesting redundant systems may act to mitigate the toxicity of nuclear-localized non-imported proteins. In

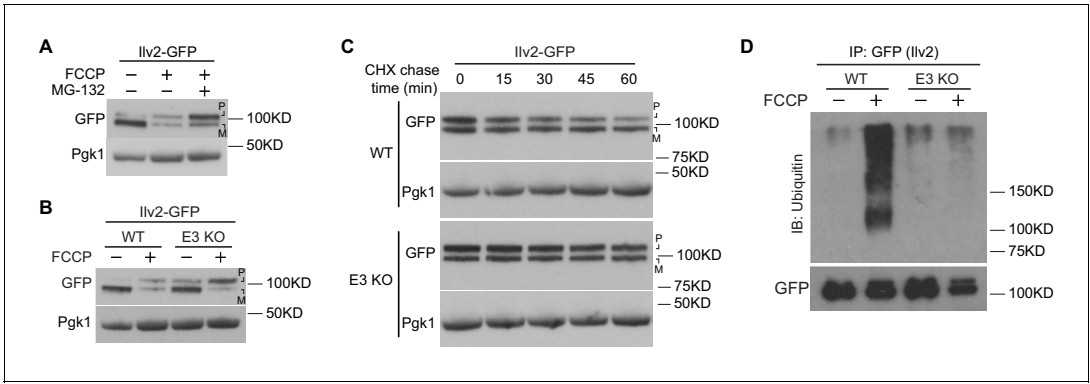

**Figure 3.** Nuclear protein quality control degrades non-imported mitochondrial proteins. (**A**) Western blot of yeast expressing Ilv2-GFP ± FCCP ± MG-132. (**B**) Western blot of yeast expressing Ilv2-GFP ± FCCP in wild-type (WT) and E3 KO strains. (**C**) Western blots showing cycloheximide (CHX) chase of Ilv2-GFP in WT and E3 KO strains in the presence of FCCP. (**D**) Western blot showing ubiquitylation of immunoprecipitated Ilv2-GFP ± FCCP in WT and E3 KO strains. Pgk1 = loading control. E3 KO = *san1Δ ubr1Δ doa10Δ*. P = precursor and M = mature.
The online version of this article includes the following figure supplement(s) for figure 3:

**Figure supplement 1.** Nuclear protein quality control promotes non-imported mitochondrial protein degradation.
**Figure supplement 2.** Nuclear protein quality control promotes non-imported mitochondrial protein degradation.

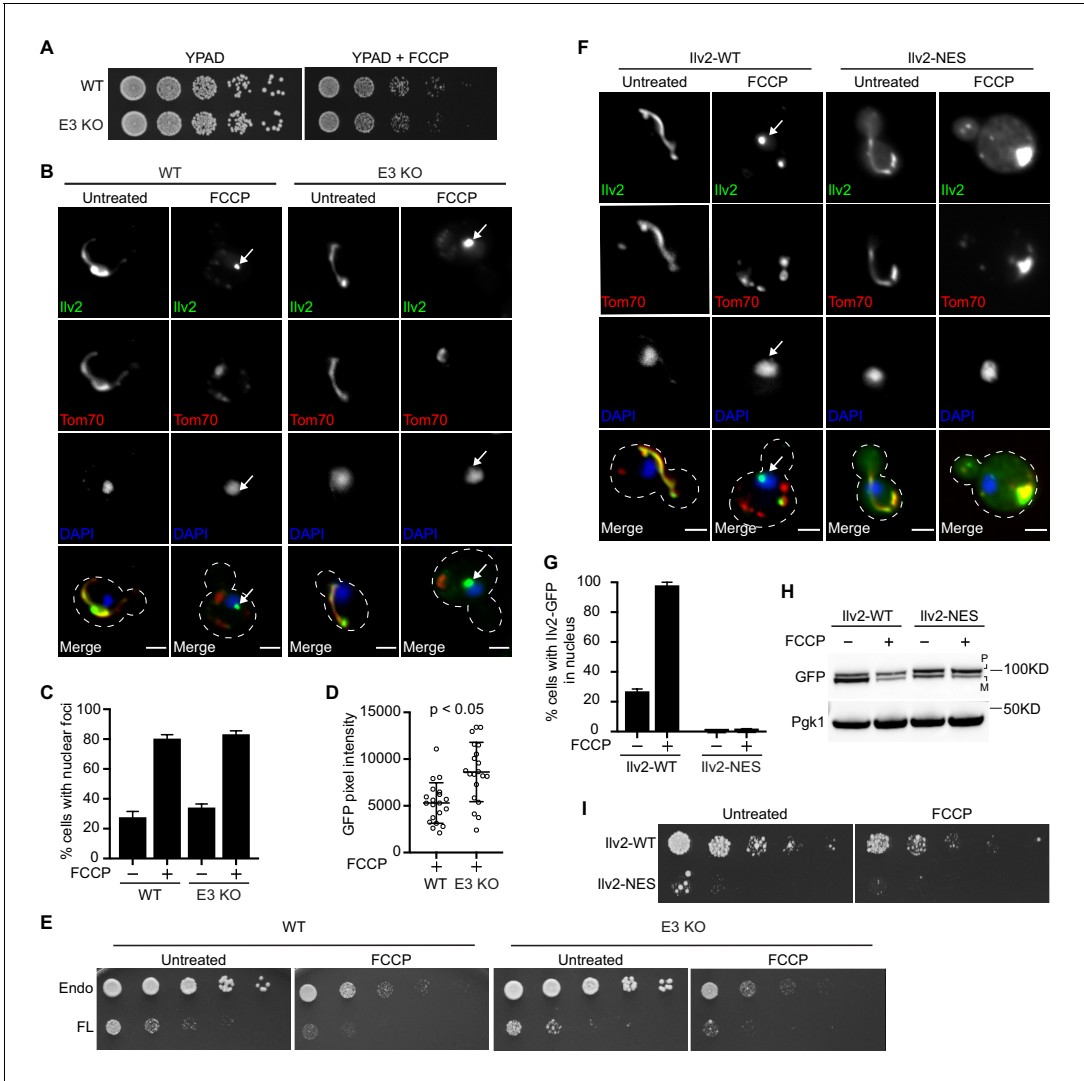

**Figure 4.** Non-imported mitochondrial precursors are toxic and localize to nuclear-associated foci when clearance is impaired. (**A**) Fivefold serial dilutions of WT and E3 KO strains on YPAD ± FCCP agar plates. (**B**) WT and E3 KO yeast expressing Ilv2-GFP and Tom70-mCherry ± FCCP. Nucleus stained with DAPI. Arrows = nuclear associated foci. Bar = 2 µm. (**C**) Quantification of (**B**). N > 99 cells per replicate, error bars = SEM of three replicates. (**D**) Quantification of average pixel intensity of Ilv2-GFP nuclear foci from (**B**). N = 20 cells, error bars = SD, p-value=0.0005. (**E**) Fivefold serial dilutions of WT and E3 KO strains expressing endogenous Ilv2-GFP (endo) ± mild overexpression of full-length Ilv2-GFP (FL) from pRS413-Ilv2-GFP on SD-His ± FCCP agar plates. (**F**) Yeast strain expressing WT Ilv2-GFP or Ilv2-NES-GFP and Tom70-mCherry ± FCCP. Nucleus stained with DAPI. Arrows = nucleus. Bar = 2 µm. (**G**) Quantification of (**F**). (**H**) Western blot of yeast expressing Ilv2-GFP or Ilv2-NES-GFP ± FCCP. (**I**) Fivefold serial dilutions of endogenously tagged WT Ilv2-GFP and Ilv2-NES-GFP strains on YPAD ± FCCP agar plates. P = precursor and M = mature.
The online version of this article includes the following figure supplement(s) for figure 4:

**Figure supplement 1.** Impaired clearance of non-imported mitochondrial proteins targets them to nuclear-associated foci.

line with that hypothesis, we noticed that in addition to diffuse nuclear localization, a portion of Ilv2-GFP accumulated in nuclear-associated foci that resembled previously described juxtanuclear (*Kaganovich et al., 2008*) or intranuclear (*Miller et al., 2015*) protein aggregate compartments (*Figure 4B*). These foci were adjacent to the DAPI-stained nucleus, excluded the mitochondrial marker Tom70-mCherry, and were present in a high percentage of FCCP-treated cells (*Figure 4C*). Prior studies showed that misfolded proteins can be sequestered into nuclear-associated aggregates when their proteasomal clearance is impaired (*Kaganovich et al., 2008*; *Miller et al., 2015*). Interestingly, we observed that the intensity of Ilv2-GFP foci increased in the E3 KO strain (*Figure 4D*). Moreover, Dld1 and Dld2, which are degraded more robustly than Ilv2, also localized to nuclear-

associated foci, but only in strains lacking the E3 ligase degradation machinery (*Figure 4—figure supplement 1B–E*).

We were unable to identify a mutation that blocked localization to these nuclear puncta, which prevented us from testing whether they act redundantly with proteasome degradation to protect cells from mitoprotein-induced toxicity. However, we did find that a twofold increase in expression of Ilv2-GFP from a single-copy plasmid resulted in constitutive localization of Ilv2-GFP to the nucleus and nuclear associated protein foci (see *Figure 5B–D*), and resulted in severe growth defects in both wild-type and E3 KO strains (*Figure 4E*). Moreover, we generated a version of Ilv2-GFP expressed from its endogenous chromosomal location containing a nuclear export signal (NES) inserted between the Ilv2 coding sequence and GFP. NES addition resulted in exclusion of Ilv2-GFP from the nucleus and lack of Ilv2-GFP turnover in the presence of FCCP (*Figure 4F–H*). Interestingly, expression of nuclear-excluded Ilv2 in cells led to severe growth defects (*Figure 4I*). Collectively, these results suggest that non-imported Ilv2 is toxic to cells and that nuclear-based proteasome destruction and aggregate sequestration may act in coordination to mitigate this toxicity.

## The MTS is required for non-imported precursor toxicity and quality control

We sought to elucidate the features of non-imported mitochondrial proteins that drive nuclear-associated aggregation, degradation, and toxicity. Mitochondrial matrix proteins such as Ilv2 are synthesized as precursors with an N-terminal MTS (*Pfanner et al., 2019*). The MTS is removed by mitochondrial-localized proteases after import (*Mossmann et al., 2012*), and failure to remove and clear MTSs leads to toxicity (*Mossmann et al., 2014*; *Poveda-Huertes et al., 2020*; *Roise et al., 1986*). To test whether the presence of an MTS on an unimported mitochondrial precursor protein is problematic, we analyzed strains containing single-copy plasmids expressing full-length Ilv2-GFP (FL), MTS-deleted Ilv2-GFP (ΔMTS), and MTS$_{Ilv2}$-GFP only (MTS) from the constitutive GPD promoter (*Figure 5A*). Like endogenous Ilv2-GFP (Endo), plasmid-derived FL-Ilv2-GFP localized to the nucleus and nuclear-associated foci, and its abundance declined with FCCP (*Figure 5B–E*). By contrast, Ilv2 lacking an MTS (ΔMTS-Ilv2-GFP) constitutively localized to the nucleus even in the absence of FCCP, but never formed nuclear-associated foci or decreased in abundance with FCCP (*Figure 5B–E*). MTS$_{Ilv2}$-GFP localized to mitochondria and exhibited no nuclear localization, puncta formation, or changes in total abundance with FCCP (*Figure 5B–E*). Thus, information in the mature, C-terminal portion of Ilv2 is necessary and sufficient for nuclear localization, but the presence of an MTS is required for non-imported Ilv2 degradation and sequestration into nuclear-associated foci.

Because Ilv2 lacking an MTS was not subjected to quality control, we wondered whether ΔMTS-Ilv2-GFP was still toxic to cells. In contrast to overexpressed FL Ilv2-GFP, which impaired growth of both wild type and E3 KO cells in the presence or absence of FCCP (*Figure 4E*), overexpressed Ilv2 lacking its MTS did not slow cell growth, and neither did overexpressed MTS$_{Ilv2}$-GFP alone (*Figure 5F*). The same was true for non-tagged versions of Ilv2 (*Figure 5—figure supplement 1A*). Thus, the presence of an MTS on unimported Ilv2 rendered the protein toxic and promoted its subsequent quality control. Importantly, the association between the presence of an MTS, degradation, and toxicity was conserved for other nuclear class proteins. Like Ilv2, Dld2 lacking its MTS constitutively localized to the nucleus, but was not subjected to degradation or sequestered into nuclear foci, and was no longer toxic to cells (*Figure 5—figure supplement 1B–F*). Moreover, degradation and toxicity of Cox15 and Lat1, which are degraded by a non-nuclear proteasome pathway, also required an N-terminal MTS (*Figure 5G,H*). Thus, the presence of an MTS on an unimported mitochondrial protein drives toxicity and targets the protein for quality control.

## MTS-mediated toxicity is not entirely linked to mitochondrial import clogging

Recent studies have shown that the failure to degrade some non-imported mitochondrial proteins leads to toxicity because the undegraded precursors clog mitochondrial import pores (*Mårtensson et al., 2019*; *Weidberg and Amon, 2018*). Since we found that the MTS of Ilv2 was required for its toxicity, we wondered whether Ilv2 overexpression and/or expression of Ilv2-NES impaired mitochondrial protein import. To test this, we first assayed the impact of full-length Ilv2 overexpression on mitochondrial protein import using a well-established in vitro import assay that

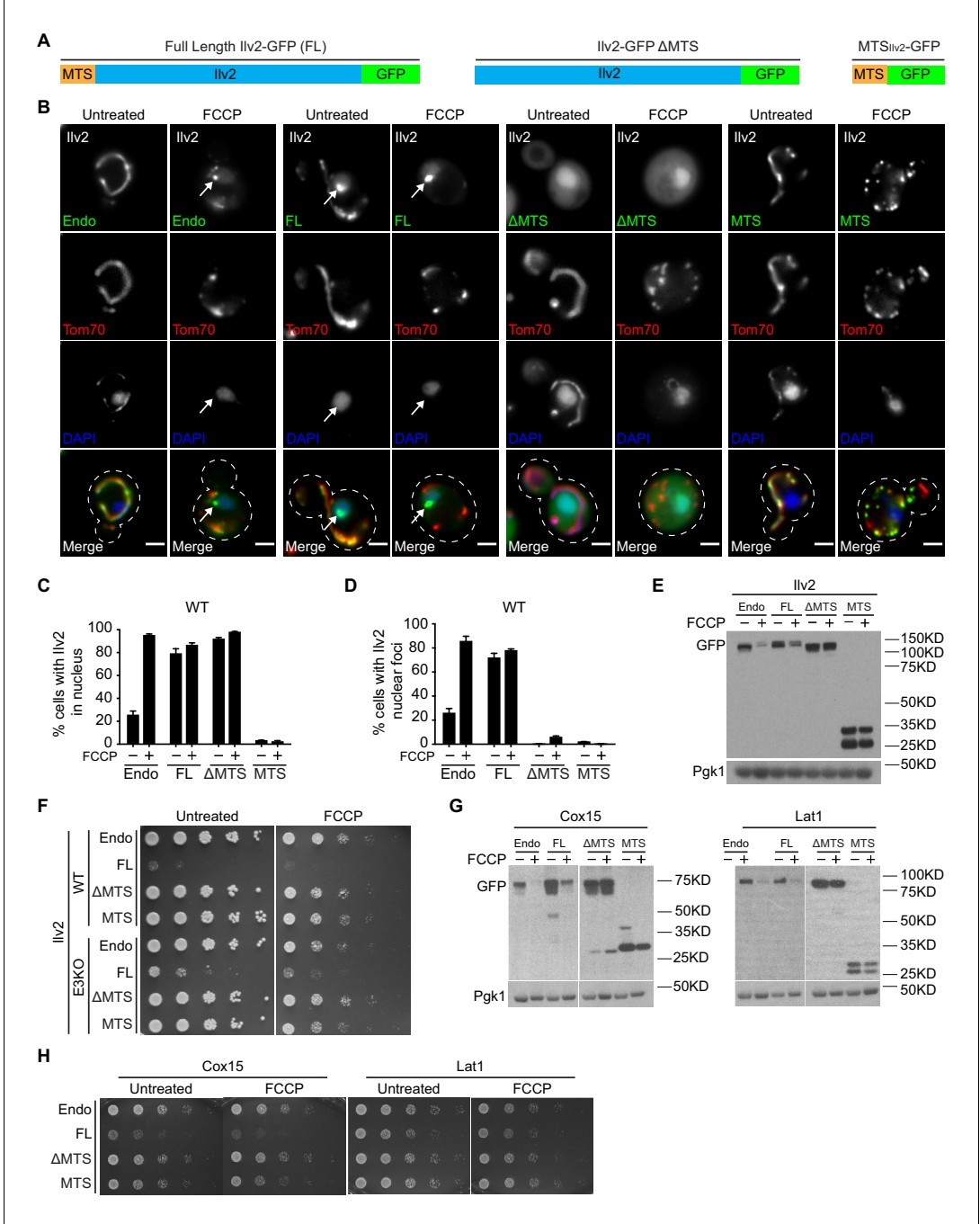

**Figure 5.** The mitochondrial targeting sequence (MTS) is required for non-imported precursor toxicity and quality control. (A) Schematic of full-length GFP-tagged Ilv2 (FL), mitochondrial targeting sequence deleted (ΔMTS) GFP-tagged Ilv2, and MTS$_{Ilv2}$ GFP only (MTS). (B) Tom70-mCherry yeast expressing endogenous Ilv2-GFP ± the indicated Ilv2 variant ± FCCP. Nucleus stained with DAPI. Arrows = nuclear-associated foci. Bars = 2 μm. (C, D) Quantification of cells with diffuse Ilv2 nuclear localization (C) or Ilv2 nuclear foci (D) from (B). For (C, D), N > 99 cells per replicate, error bars = SEM of three replicates. (E) Western blot of strains expressing indicated Ilv2-GFP variants ± FCCP. Pgk1 = loading control. (F) Fivefold serial dilutions of WT and E3 KO strains expressing endogenous Ilv2-GFP (endo) ± mild overexpression of the indicated Ilv2-GFP variants on SD-His ± FCCP agar plates. (G) Western blot of strains expressing indicated Cox15-GFP or Lat1-GFP variants, respectively, ± FCCP. Pgk1 = loading control. (H) Fivefold serial dilutions of WT strains expressing endogenous Cox15-GFP or Lat1-GFP (endo) ± mild overexpression of the indicated variants on SD-His ± FCCP agar plates. P = precursor and M = mature.

The online version of this article includes the following figure supplement(s) for figure 5:

**Figure supplement 1.** The MTS is required for non-imported precursor toxicity and degradation.

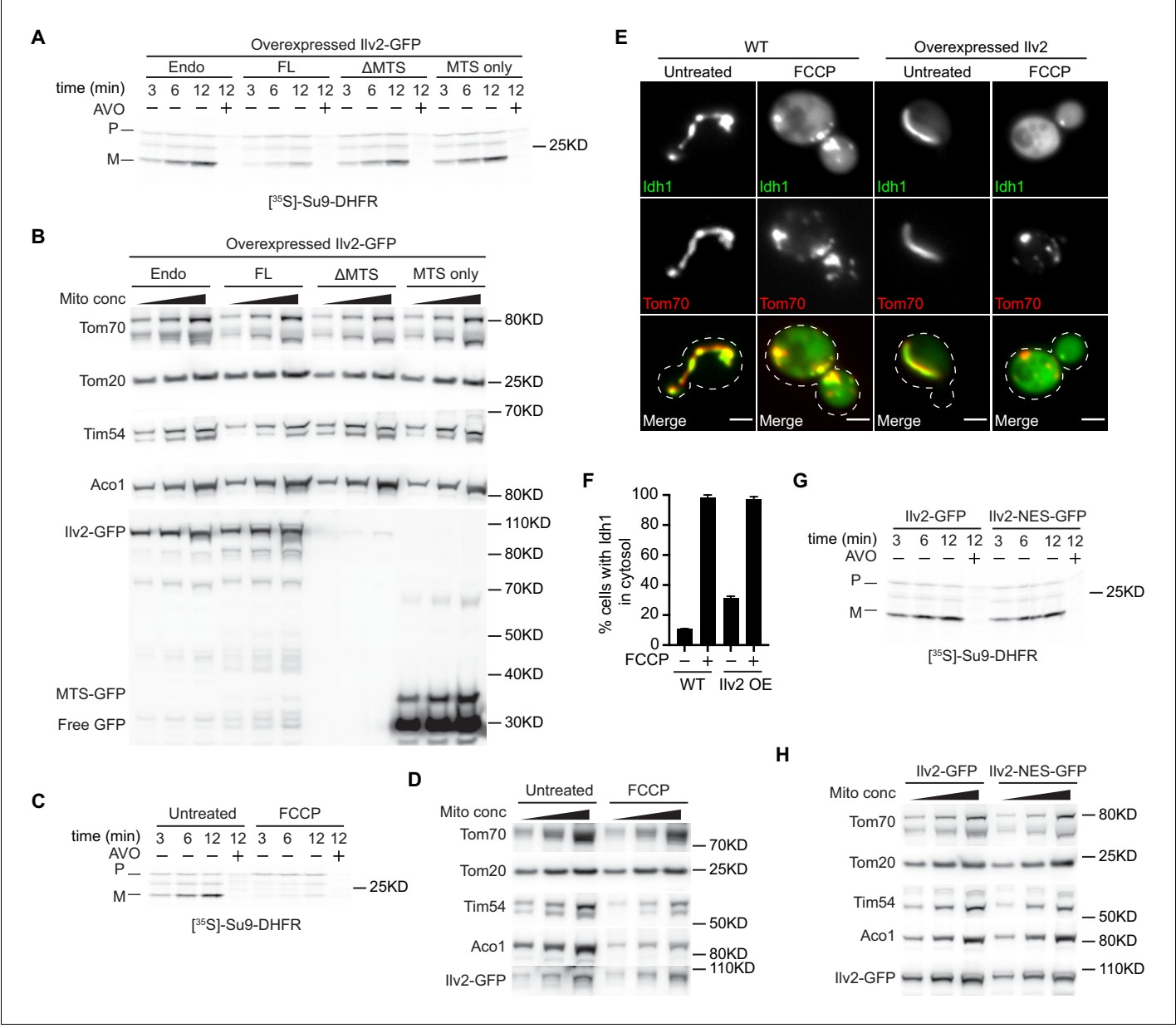

**Figure 6.** MTS-mediated toxicity of Ilv2 is not entirely linked to mitochondrial import clogging. (**A**) [35]S-labeled Su9-DHFR was imported into mitochondria isolated from yeast overexpressing the indicated Ilv2-GFP variants. (**B**) Western blot showing protein levels in mitochondria isolated from yeast overexpressing the indicated Ilv2-GFP variants. (**C**) [35]S-labeled Su9-DHFR was imported into mitochondria isolated from yeast expressing endogenous Ilv2-GFP ± FCCP. (**D**) Western blot showing protein levels in mitochondria isolated from yeast expressing endogenous Ilv2-GFP ± FCCP. (**E**) Yeast expressing Idh1-GFP and Tom70-mCherry in WT or Ilv2 (untagged) overexpressing strains ± FCCP. Bars = 2 µm. (**F**) Quantification of (**E**). (**G**) [35]S-labeled Su9-DHFR was imported into mitochondria isolated from yeast expressing endogenous Ilv2-GFP or Ilv2-NES-GFP ± FCCP. (**H**) Western blot showing protein levels in mitochondria isolated from yeast expressing endogenous Ilv2-GFP or Ilv2-NES-GFP. For (**A**), (**C**), and (**G**), a mixture of antimycin A, oligomycin, and valinomycin (AVO) was used to dissipate the membrane potential. Non-imported proteins were proteolytically removed with proteinase K and the import was analyzed by SDS–PAGE and autoradiography. P = precursor and M = mature.

analyzes uptake and subsequent processing of in vitro translated DHFR fused to the MTS of *Neurospora crassa* ATP synthase subunit 9 (Su9-DHFR) (*Pfanner et al., 1987*). We found that overexpression of GFP-tagged full-length Ilv2 (FL), but not MTS-deleted Ilv2-GFP (ΔMTS) or MTS$_{Ilv2}$ only (MTS)-GFP, decreased import of Su9-DHFR into isolated mitochondria, as indicated by reduced formation of the mature form of Su9-DHFR during incubation (*Figure 6A*). Consistent with these effects, we also found that mitochondria isolated from strains overexpressing FL-Ilv2-GFP contained lower

steady-state levels of mitochondrial IM protein Tim54, but not the OM proteins Tom70 and Tom20 (*Figure 6B*). The reduction in the import of Su9-DHFR and steady-state abundance of mitochondrial proteins in strains overexpressing Ilv2-GFP was not as strong as those observed in mitochondria isolated from cells treated with FCCP (*Figure 6C,D*), suggesting the impact of Ilv2 overexpression on mitochondrial import is not as severe as loss of mitochondrial membrane potential. Consistent with this observation, Ilv2 overexpression led to the cytoplasmic accumulation of the mitochondrial precursor Idh1-GFP in over 30% of cells, as compared to nearly 100% cytoplasmic accumulation observed in cells treated with FCCP (*Figure 6E,F*, and as reported in screen results reported in *Figure 2*). Together, these results indicate that overexpressed full-length Ilv2-GFP partially blocks mitochondrial import, but not to the same extent as the mitochondrial depolarizing agent FCCP.

We also tested the impact of Ilv2-NES-GFP on mitochondrial import. Ilv2-NES-GFP is a version of Ilv2 that is expressed from its native promoter and is constitutively excluded from the nucleus and highly toxic to cells (*Figure 4F and I*). Remarkably, we found that Ilv2-NES-GFP, despite its high toxicity, had little to no impact on the in vitro import of Su9-DHFR and had only mild effects on the steady-state levels of Tim54 and Aco1 in isolated mitochondria (*Figure 6G,H*). Given the fact that this version of Ilv2 is highly toxic to cells both in the presence and in the absence of FCCP, it seems that the toxicity of non-imported Ilv2 cannot be completely attributed to its ability to act as a mitochondrial import clogger, and likely stems from additional deleterious effects of the protein on unidentified cellular processes.

## Discussion

Prior studies demonstrated that the accumulation of unimported mitochondrial precursors causes proteotoxicity termed mitoprotein-induced stress (*Wang and Chen, 2015*; *Wrobel et al., 2015*). To combat this stress, cells mount a coordinated response that involves upregulation of proteasome capacity (*Boos et al., 2019*; *Wrobel et al., 2015*), downregulation of translation (*Wang and Chen, 2015*), and clearance of precursors that accumulate at the mitochondrial surface (*Mårtensson et al., 2019*; *Metzger et al., 2020*; *Weidberg and Amon, 2018*) and ER membrane (*Hansen et al., 2018*; *Laborenz et al., 2021*). Here, we surveyed the mitochondrial proteome under conditions of mitochondrial import impairment to get a clearer picture of the full spectrum of fates for non-imported mitochondrial proteins. We found that mitochondrial precursors accumulate in many regions of the cell and identified the nucleus as an important quality control destination for sequestering and destroying non-imported mitochondrial proteins. Moreover, we demonstrated that the N-terminal MTS is a major driver of non-imported protein toxicity. Our findings also indicate that non-imported mitochondrial proteins represent a large class of potential substrates for nuclear protein quality control. This discovery raises the intriguing possibility that non-imported mitochondrial proteins may synergize with other aggregate-prone proteins to overwhelm protein quality control systems during aging and disease (*Park et al., 2013*). Future studies to determine what drives non-imported mitochondrial proteins to various cellular destinations, and addressing whether mitoproteins perform signaling or metabolic functions at these sites (much like the mito-nuclear signaling protein ATFS1) (*Nargund et al., 2012*) will help illuminate how cells adapt to the significant burden of mitochondrial dysfunction.

## Materials and methods

### Key resources table

| Reagent type (species) or resource | Designation | Source or reference | Identifiers | Additional information |
|---|---|---|---|---|
| Strain, strain background *Saccharomyces cerevisiae* | Yeast GFP collection co-expressing Tom70-mCherry | PMID:27097106 | N/A | N/A |
| Strain, strain background (*S. cerevisiae*) | BY4741 | ATCC | Cat # 201388 | N/A |
| Strain, strain background (*S. cerevisiae*) | BY4743 | ATCC | Cat # 201390 | N/A |

*Continued on next page*

*Continued*

| Reagent type (species) or resource | Designation | Source or reference | Identifiers | Additional information |
|---|---|---|---|---|
| Strain, strain background (*S. cerevisiae*) | Other strains used in this study | This paper | *Supplementary file 2* | N/A |
| Antibody | Anti-GFP (mouse monoclonal) | Millipore Sigma | Cat # 11814460001 RRID: AB_390913 | WB (1:2500) |
| Antibody | Anti-HA (mouse monoclonal) | Millipore Sigma | Cat # 11583816001 RRID: AB_514506 | WB (1:1000), IF (1:200) |
| Antibody | Anti-PGK1 (mouse monoclonal) | Abcam | Cat # 22C5D8 RRID: AB_2756444 | WB (1:1000) |
| Antibody | Anti-FLAG M2 (mouse monoclonal) | Millipore Sigma | Cat # F1804 RRID: AB_262044 | IF (1:200) |
| Antibody | Anti-H2B (rabbit polyclonal) | Active Motif | Cat # 39947 RRID: AB_2793403 | WB (1:1000) |
| Antibody | Anti-Idh1 (goat polyclonal) | ORIGENE | Cat # Ap31099PU-N | WB (1:500) |
| Antibody | Anti-Ilv2 (rabbit polyclonal) | Dr. Agnieszka Chacinska | N/A | WB (1:1000), IF (1:200) |
| Antibody | Anti-Tom70 (rabbit polyclonal) | Dr. Nikolaus Pfanner | N/A | WB (1:1000) |
| Antibody | Anti-Tom20 (rabbit polyclonal) | Dr. Nikolaus Pfanner | N/A | WB (1:1000), IF (1:200) |
| Antibody | Anti-Tim54 (rabbit polyclonal) | Dr. Nikolaus Pfanner | N/A | WB (1:1000) |
| Antibody | Anti-Tim50 (rabbit polyclonal) | Dr. Nikolaus Pfanner | N/A | WB (1:1000) |
| Antibody | Anti-Tim18 (rabbit polyclonal) | Dr. Nikolaus Pfanner | N/A | WB (1:1000) |
| Antibody | Anti-OM45 (rabbit polyclonal) | Dr. Carla Koehler | N/A | WB (1:1000), IF (1:200) |
| Antibody | anti-Tom40 (rabbit polyclonal) | Dr. Toshiya Endo | N/A | WB (1:1000) |
| Recombinant DNA reagent | pKT128 (plasmid) | Addgene | Plasmid # 8729 | N/A |
| Recombinant DNA reagent | pKT127-mCherry (plasmid) | Daniel Gottschling | N/A | N/A |
| Recombinant DNA reagent | pFA6a-5FLAG-KanMX6 (plasmid) | Addgene | Plasmid # 15983 | N/A |
| Recombinant DNA reagent | pFA6A-mCherry-HphMX (plasmid) | Addgene | Plasmid # 105156 | N/A |
| Recombinant DNA reagent | pVL015 (plasmid) | Addgene | Plasmid # 64766 | N/A |
| Recombinant DNA reagent | pFA6A-3HA-KanMX (plasmid) | Addgene | Plasmid # 39295 | N/A |
| Recombinant DNA reagent | Template plasmid for $^{35}$S-radiolabeled Su9-DHFR | Walter Neupert PMID:2884042 | N/A | N/A |
| Recombinant DNA reagent | Other plasmids generated in this study | This paper | *Supplementary file 4* | N/A |
| Sequenced-based reagent | Oligos used in this study | This paper | *Supplementary file 3* | N/A |
| Chemical compound, drug | $\beta$-Estradiol | Millipore Sigma | Cat # E8875; CAS # 50-28-2 | N/A |
| Chemical compound, drug | Carbonyl cyanide 4-(trifluoromethoxy) phenylhydrazone (FCCP) | Millipore Sigma | Cat # C2920; CAS # 370-86-5 | N/A |
| Chemical compound, drug | cOmplete Protease Inhibitor Cocktail | Millipore Sigma | Cat# 11697498001 | N/A |
| Chemical compound, drug | Dimethyl sulfoxide | Millipore Sigma | Cat # D2650; CAS # 67-68-5 | N/A |
| Chemical compound, drug | Cycloheximide | Millipore Sigma | Cat # C1988; CAS # 66-81-9 | N/A |

*Continued on next page*

*Continued*

| Reagent type (species) or resource | Designation | Source or reference | Identifiers | Additional information |
|---|---|---|---|---|
| Chemical compound, drug | Doxycycline hyclate | Millipore Sigma | Cat # C9891; CAS # 24390-14-5 | N/A |
| Chemical compound, drug | Polyvinylpyrrolidone | Millipore Sigma | Cat # PVP40; CAS # 9003-39-8 | N/A |
| Chemical compound, drug | Pepstatin | Millipore Sigma | Cat # 10253286001; CAS # 26305-03-3 | N/A |
| Chemical compound, drug | Phenylmethylsulfonyl fluoride | Millipore Sigma | Cat # P7626; CAS # 329-98-6 | N/A |
| Chemical compound, drug | Calcofluor Fluorescent Brightener 28 | Millipore Sigma | Cat # F3543; CAS # 4404-43-7 | N/A |
| Chemical compound, drug | Diamidino-2-phenylindole dihydrochloride (DAPI) | Thermo Fisher | Cat # D1306 | N/A |
| Chemical compound, drug | Glass Antifade Mountant with NucBlue Stain (P36981) | Thermo Fisher | Cat # P36981 | N/A |
| Chemical compound, drug | (S)-MG-132 | Cayman Chemical | Cat # 10012628; CAS # 133407-82-6 | N/A |
| Chemical compound, drug | N-ethylmaleimide (NEM) | Sigma–Aldrich | Cat # S3876; CAS# 128-53-0 | N/A |
| Chemical compound, drug | IGEPAL NP-40-substitute | Sigma–Aldrich | Cat # CA-630; CAS # 9002-93-1 | N/A |
| Chemical compound, drug | Zymolyase 100T | US Biological Life Sciences | Cat # Z1004; CAS# 37340-57-1 | N/A |
| Chemical compound, drug | Triton X-100 | Bio-Rad | Cat # 1610407; CAS# 9002-93-1 | N/A |
| Chemical compound, drug | Formaldehyde 16% in aqueous solution, EM Grade | VWR | Cat # 100503–914; CAS# 50-00-0 | N/A |
| Chemical compound, drug | DTT (dithiothreitol) (>99% pure) protease free | GOLDBIO | Cat # DTT10; CAS# 27565-41−9/3483- 12−3 | N/A |
| Chemical compound, drug | Antimycin A | Sigma–Aldrich | Cat # A8674; CAS# 1397-94-0 | N/A |
| Chemical compound, drug | Phusion polymerase | New England Biolabs | Cat # M0530L | N/A |
| Chemical compound, drug | Valinomycin | Sigma–Aldrich | Cat # V0627; CAS# 2001-95-8 | N/A |
| Chemical compound, drug | Oligomycin | Sigma–Aldrich | Cat # 75351; CAS# 579-13-5 | N/A |
| Commercial assay or kit | Gibson Assembly Cloning Kit | New England Biolabs | Cat # E5510S | N/A |
| Commercial assay or kit | TNT SP6 Quick Coupled Transcription/Translation System | Promega | Cat # L2080 | N/A |
| Software, Algorithm | FIJI | PMID:22743772 | Version 1 | N/A |
| Software, Algorithm | Prism | Graphpad Software | Version 8 | N/A |
| Software, Algorithm | SnapGene | GSL Biotech | Version 4.2 | N/A |
| Software, Algorithm | Image Lab | Bio-Rad | Version 6 | N/A |
| Software, Algorithm | Zen Blue | Carl Zeiss | Version 2.6 | N/A |
| Software, Algorithm | Zen Black | Carl Zeiss | Version 2.3 | N/A |
| Software, Algorithm | Mitoprot | PMID:8944766 | N/A | N/A |
| Software, Algorithm | Illustrator CC | Adobe | Version 22.1 | N/A |
| Software, Algorithm | Photoshop CC | Adobe | Version 19.1 | N/A |

## Reagents

Chemicals were obtained from the following sources: $\beta$-estradiol (E8875), carbonyl cyanide 4-(tri-fluoromethoxy) phenylhydrazone (C2920), cOmplete Protease Inhibitor Cocktail (11697498001), dimethyl sulfoxide (D2650), cycloheximide (C1988), doxycycline hyclate (C9891), polyvinylpyrrolidone (PVP40), pepstatin (10253286001), phenylmethylsulfonyl fluoride (P7626), calcofluor Fluorescent Brightener 28 (F3543) from Millipore Sigma, 4′,6-diamidino-2-phenylindole dihydrochloride (DAPI) (D130), ProLong Glass Antifade Mountant with NucBlue Stain (P36981) from ThermoFisher, (S)-MG-132 (10012628) from Cayman Chemical, N-ethylmaleimide (NEM) (S3876), IGEPAL NP-40 (CA-630) from Sigma–Aldrich, Zymolyase 100T (Z1004) from US Biological Life Sciences, Triton X-100 (1610407) from Bio-Rad, paraformaldehyde (100503–914) from VWR, and dithiothreitol (DTT10) from GOLDBIO. Antibodies and other reagents are described in the appropriate section below.

## Yeast strains

All yeast strains are derivatives of *S. cerevisiae* S288c (BY) (*Brachmann et al., 1998*) and are listed in *Supplementary file 2*. Strains expressing fluorescently tagged proteins from their native loci were created by one-step PCR-mediated C-terminal endogenous epitope tagging using standard techniques and the oligo pairs listed in *Supplementary file 3* (*Brachmann et al., 1998*; *Sheff and Thorn, 2004*). Plasmid templates for GFP and mCherry tagging were from the pKT series of vectors (*Sheff and Thorn, 2004*), plasmid template for RITE tagging was previously described pVL015 (*Verzijlbergen et al., 2010*), and plasmid templates for FLAG, HA, and mCherry tagging were pFA6A-5FLAG-KanMX (Addgene 15983) (*Noguchi et al., 2008*), pFA6A-3HA-His3MX (Addgene 41600) (*Longtine et al., 1998*), pFA6A-3HA-KanMX (Addgene 39295) (*Bähler et al., 1998*), and pFA6A-mCherry-HphMX (Addgene 105156) (*Wang et al., 2014*). Deletion strains were created by one-step PCR-mediated gene replacement using the oligos pairs listed in *Supplementary file 3* and plasmid templates of pRS series vectors (*Brachmann et al., 1998*). Correct integrations were confirmed with a combination of colony PCR across the chromosomal insertion site and correctly localized expression of the fluorophore by microscopy. The strain collection used for screening in *Figure 2* expressed Tom70-mCherry/any protein-GFP and was created previously (*Hughes et al., 2016*). The genotype of all strains in the collection is MATa/MATα his3Δ1/his3Δ1 leu20/leu2Δ0 ura3Δ0/ura3Δ0 met15Δ0/+ lys2Δ0/+ anygene-GFP-His3MX/+ TOM70-mCherry-KanMX/+.

## Yeast cell culture and media

For all microscopy and western blot experiments, yeast were grown exponentially for 15 hr up to a maximum density of $1 \times 10^7$ cells/mL prior to starting any treatments. Cells were cultured as indicated in the main text and figure legends in YPAD medium (1% yeast extract, 2% peptone, 0.005% adenine, 2% glucose) or synthetic defined medium lacking histidine (SD-His) (0.67% yeast nitrogen base without amino acids, 2% glucose, supplemented nutrients 0.074 g/L each adenine, alanine, arginine, asparagine, aspartic acid, cysteine, glutamic acid, glutamine, glycine, myo-inositol, isoleucine, lysine, methionine, phenylalanine, proline, serine, threonine, tryptophan, tyrosine, uracil, valine, 0.369 g/L leucine, 0.007 g/L para-aminobenzoic acid). FCCP and MG-132 were used at a final concentration of 10 µM and 50 nM, respectively. All FCCP and/or MG-132 treatments were conducted for 6 hr. For knockdown of TOM40 expressed under control of the tetracycline promoter, cultures were grown in log-phase for 16 hr in the presence of doxycycline (20 µg/mL) prior to any experimental treatments. The wild-type control strain was cultured under the same conditions. For RITE tag-switching experiments, β-estradiol was added to cultures at a final concentration of 1 µM to induce tag switching. FCCP was added to cultures at a final concentration of 10 µM at the same time of β-estradiol. Cultures were imaged after 6 hr of treatment.

## Plasmids and cloning

Centromeric yeast plasmids expressing GPD-promoter-driven full-length, MTS-deleted, or MTS-only versions of untagged Ilv2 and full-length, MTS-deleted, or MTS-only variants of Ilv2, Lat1, Cox15, and Dld2 fused to C-terminal GFP epitopes were assembled using Gibson Assembly Master Mix (E2611L, NEB) following the manufacturer's instructions. Plasmid names and construction details (including PCR templates, oligo pairs, and digested plasmid templates) used in Gibson Assembly are listed in *Supplementary file 4*. PCR amplifications from yeast genomic DNA and plasmid DNA were

conducted with Phusion Polymerase (M0530L, NEB) using oligonucleotides listed in *Supplementary file 3*. Plasmids were verified by sequencing.

## MTS prediction

MTSs for Ilv2, Cox15, Lat1, and Dld2 were predicted using Mitoprot (*Claros and Vincens, 1996*). Correct MTS prediction was confirmed by analyzing localization of C-terminal GFP-tagged versions of MTS-only or MTS-deleted proteins via microscopy.

## Microscopy

Two hundred to 300 nm optical Z-sections of live yeast cells were acquired with an AxioImager M2 (Carl Zeiss) equipped with an Axiocam 506 monochromatic camera (Carl Zeiss) and 100× oil-immersion objective (Carl Zeiss, Plan Apochromat, NA 1.4) or with an AxioObserver 7 (Carl Zeiss) equipped with a PCO Edge 4.2LT Monochrome, Air Cooled, USB 3 CCD camera with a Solid-State Colibri 7 LED illuminator and 100× oil-immersion objective (Carl Zeiss, Plan Apochromat, NA 1.4). All images were acquired with ZEN (Carl Zeiss) and processed with Fiji (NIH). All images shown in figures represent a single optical section.

## DAPI staining

Yeast cells were stained with DAPI by incubating cultures for 10 min in respective growth media with DAPI (1 µg/mL).

## Quantification of nuclear-associated foci intensity

Mean GFP pixel intensity of nucleus and nuclear-associated foci was calculated via line scan analysis of pixel intensity from maximum-intensity projections on 20 cells using FIJI (NIH) (*Schindelin et al., 2012*). Nucleus stained by DAPI was used as a reference to draw lines of ~2.5 µm for analysis.

## Determination of replicative age

Yeast strains exponentially growing for 15 hr up to a maximum density of $1 \times 10^7$ cells/mL were stained with for 5 min in YPAD with 5 µg/mL of Fluorescent Brightener 28 (F3543, Millipore Sigma), which stains bud scars. The replicative age of each yeast cell was determined by counting of the number of bud scars after staining. Cells with less than five bud scars were categorized as young and cells with five or more bud scars were categorized as old.

## Indirect immunofluorescence staining

For indirect immunofluorescence staining, cells were harvested by centrifugation and fixed in 10 mL fixation medium (4% paraformaldehyde in YPAD) for 1 hr. Fixed yeast cells were washed with Wash Buffer (0.1 M Tris, pH = 8, 1.2 M sorbitol) twice and incubated with DTT (10 mM DTT in 0.1M Tris, pH = 9.4) for 10 min. Spheroplasts were generated by incubating cells in solution containing 0.1 M KPi, pH = 6.5, 1.2 M sorbitol, and 0.25 mg/mL Zymolyase at 30°C for 30 min. Spheroplasts were gently diluted in 1:40 using Wash Buffer and attached to glass slides pre-coated with 0.1% poly-L-Lysine (2 mg/mL). Samples were permeabilized in cold 0.1% Triton-X100 in PBS for 10 min at 4°C, briefly dried and blocked (30 min at room temperature) in Wash Buffer containing 1% bovine serum albumin (BSA). After blocking, samples were incubated with 1:200 diluted primary antibodies against FLAG (F1804, Millipore Sigma), Ilv2, Tom20, or OM45 for 90 min followed by washing 10 times. Samples were then incubated with 1:300 diluted secondary antibody (A32723, Invitrogen) followed by washing 10 times. Antibody dilutions were made using Wash Buffer containing 1% BSA. Samples were washed with Wash Buffer containing 1% BSA and 0.1% Tween-20. Slides were washed twice with Wash Buffer before sealing and mounted with hardset medium containing NucBlue stain (P36981, Invitrogen) overnight. Widefield images were acquired as described above in microscopy section.

## Protein preparation and western blotting

Western blotting of yeast extracts was carried out as described previously (*Hughes et al., 2016*). Briefly, $1 \times 10^7$ log phase yeast cells were harvested and resuspended in 50 µL of $H_2O$. Fifty microliters of NaOH (1 M) was added to cell suspension and incubated for 5 min at room

temperature. Cells were centrifuged at 20,000 × g for 10 min at 4°C, and cell pellets were resuspended in sodium dodecyl sulfate (SDS) lysis buffer (30 mM Tris–HCl pH 6.8, 3% SDS, 5% glycerol, 0.004% bromophenol blue, 2.5% β-mercaptoethanol). Cells extracts were resolved on Bolt 4–12% Bis–Tris Plus Gels (NW04125BOX, Thermo Fisher) with NuPAGE MES SDS Running Buffer (NP0002-02, Thermo Fisher) and transferred to nitrocellulose membranes. Membranes were blocked and probed in blocking buffer (1× PBS, 0.05% Tween 20, 5% non-fat dry milk) using the primary antibodies for GFP (1814460001, Sigma Millipore), HA (11583816001, Sigma Millipore), Ilv2 (gift from Dr. Agnieszka Chacinska, International Institute of Molecular and Cell Biology), Tom70, Tom20, Tim54, Tim50, Tim18 (gifts from Dr. Nikolaus Pfanner, University of Freiburg), OM45 (gift from Dr. Carla Koehlar, UCLA), Idh1 (Ap31099PU-N, ORIGENE) or Pgk1 (22C5D8, Abcam), and HRP-conjugated secondary antibodies (715-035-150, Jackson Immunoresearch). Blots were developed with SuperSignal West Pico Chemiluminescent substrate (34580, Thermo Fisher) and exposed to films. Blots were developed using film processor (SRX101, Konica Minolta) or a Chemidoc MP system (Bio-Rad).

## Nuclear enrichment

Cells were grown in log-phase overnight as described above followed by treatment with MG-132 and ± FCCP for 4 hr. $4 \times 10^8$ total cells were harvested. Cells were washed with ddH$_2$O, and the wet weight of the pellet was recorded. Cells were incubated in DTT Buffer (100 mM Tris–HCl pH 9.5, 10 mM DTT) and 50 nM MG-132 with gentle shaking at 30°C for 20 min. Cells were then spheroplasted via incubation in zymolyase buffer (1.2 M sorbitol, 20 mM K$_2$HPO$_4$, pH 7.4), 50 nM MG-132, and 1 mg of Zymolyase 100T (Z1004, US Biological Life Sciences) per 1 g cell pellet for 1 hr at 30°C with gentle shaking. Spheroplasts were washed once with zymolyase buffer, and then all subsequent steps were carried out on ice. Spheroplasts were dounce-homogenized with 35 strokes in 5 mL of polyvinylpyrrolidone-40 solution (8% PVP-40, 20 mM K-phosphate, 7.5 μM MgCl$_2$, pH 6.5), 0.025% Triton X-100, 5 mM DTT, 50 μL Solution P (20 mg/mL phenylmethylsulfonyl fluoride (PMSF), 0.4 mg/mL Pepstatin A in ethanol), and 50 μL 100× cOmplete protease inhibitor cocktail 697498001, Millipore Sigma). Next, 15 mL of PVP-40 solution, 15 μL Solution P, and 15 μL PIC were added, and spheroplasts were dounce-homogenized with an additional five strokes. PVP-40 ensures nuclei stay intact during lysis (*Niepel M et al., 2017*). The cell lysate was centrifuged for 3000 × g for 5 min. The resulting supernatant was discarded, and pellets were washed once and resuspended in 1 mL of IP Buffer (50 mM Tris pH7.5, 150 mM NaCl, 1 mM ethylenediaminetetraacetic acid (EDTA), 10% glycerol, 1% IGEPAL [NP-40 substitute], 100 μM PMSF). Intact nuclei, which are more resistant to NP-40 than other cellular membranes, were immobilized non-specifically to magnetic agarose beads (BMAB 20, Chromotek) via incubation at 4°C for 2–3 hr. After binding, nuclei were washed 4 × 15 min in IP buffer at 4°C. Nuclear-enriched extracts were eluted by incubating beads in 2× Laemmli buffer (63 mM Tris pH 6.8, 2% [w/v] SDS, 10% [v/v] glycerol, 1 mg/mL bromophenol blue, 1% [v/v] b-mercaptoethanol) at 90°C for 10 min. Eluates were subjected to sodium dodecyl sulfate–polyacrylamide gel electrophoresis (SDS–PAGE) and western blotting with primary anti-HA antibody (11583816001, Sigma Millipore), anti-Tom70 and Tim44 antisera (gifts from Dr. Nikolaus Pfanner, University of Freiburg), anti-GFP antibody (1814460001, Millipore Sigma), and anti-H2b antibody (39947, Active Motif). Effectiveness of nuclear enrichment was indicated by increase in relative abundance of nuclear markers H2B and Nup49-GFP and decrease in Tom70 and Tim44 in nuclear extracts compared to whole-cell lysate. Nuclei were monitored during isolation by visualizing Nup49-GFP via fluorescence microscopy.

## Cycloheximide-chase analysis

Exponentially growing cells were treated ± FCCP for 4 hr, after which, cycloheximide (100 μg/mL) was added to the cultures. The time zero sample was collected immediately after adding cycloheximide. For all other timepoints, samples were collected by harvesting an equal volume of media to that which was harvested at time zero. Samples were then subjected to SDS–PAGE and western blotting with primary antibodies for HA (11583816001, Sigma Millipore) or GFP (1814460001, Sigma Millipore) and Pgk1 (22C5D8, Abcam). Blots were developed as described above.

## Microscopy and western blot screens

Individual strains listed in *Supplementary file 1* from the Tom70-mCherry/mitochondrial protein GFP collection were cultured in batches overnight in YPAD as described above and then incubated ± FCCP for 6 hr. After treatment, cultures were split for simultaneous microscopy and western blot analysis. Images and western blots were analyzed and scored by three independent researchers. A subset of strains from each class was reconstructed and reanalyzed with both FCCP and genetic ablation of mitochondrial import. Class assignments were based on combined results of microscopy and western blot analysis and were as follows: class 1 (nucleus), small to large decrease in protein levels and localized to the nucleus in the presence of FCCP; class 2 (mitochondria), minimal change in protein level and robustly localized to mitochondria in the presence of FCCP; class 3 (cytoplasm), no change or an increase in protein level and localized predominantly to the cytoplasm with FCCP treatment; class 4 (ER), mild or no change in protein abundance and localized to ER upon FCCP; and class 5 (reduced abundance), large reduction in protein abundance and no longer easily detectable via microscopy with FCCP treatment.

## Immunoprecipitation

Cells were grown as described above and treated ± FCCP and MG-132 for 6 hr. $1 \times 10^8$ total cells were harvested, resuspended in 1 mL of lysis buffer (50 mM Tris pH7.5, 150 mM NaCl, 1 mM EDTA, 10% glycerol, 1% IGEPAL [NP-40 substitute], 100 µM PMSF, and 10 mM NEM), and lysed with glass beads using an Omni Bead Ruptor 12 Homogenizer (eight cycles of 20 s each). Cells lysates were cleared by centrifugation at 20,000 × g, and supernatant was moved to a new tube. Cell pellets were resuspended in 50 µL of SUME buffer (1% SDS, 8 M urea, 10 mM MOPS, pH 6.8, 10 mM EDTA, and 10 mM NEM) and heated at 55°C for 5 min. Fifty microliters of cell pellet resuspension was combined with supernatant from lysate clearance centrifugation, and total volume was adjusted to 1 mL by adding lysis buffer. Lysates were incubated with 25 µL of anti-GFP bead slurry (GTMA, GFP-Trap_MA, chromotek) at 4°C for 3–4 hr and then washed 4× for 10 min each in lysis buffer (without NEM). Immunoprecipitated proteins were eluted by incubating beads in 2× Laemmli buffer (63 mM Tris pH 6.8, 2% [w/v] SDS, 10% [v/v] glycerol, 1 mg/mL bromophenol blue, 1% [v/v] β-mercaptoethanol) at 90°C for 10 min. Eluates were subjected to SDS–PAGE and western blotting with primary anti-ubiquitin antibody (PA1-187, ThermoFisher) and anti-GFP antibody (1814460001, Sigma Millipore). Blots were developed as described above.

## Growth assays

Fivefold serial dilutions of exponentially growing yeast cells were diluted in YPAD or SD-His, and 3 µL of each dilution was spotted onto YPAD or SD-His media plates with or without FCCP (5 µM). Approximate number of cells plated at each dilution spot was 5000, 1000, 20, 40, and 8. Plates were incubated at 30°C for 24–48 hr before obtaining images.

## Mitochondria isolation

Yeast cells were grown as described above until midlogarithmic growth phase ($OD_{600} \approx 1$), isolated by centrifugation, washed with $dH_2O$, and the pellet weight was determined. Subsequently, cells were resuspended in 2 mL/g pellet DTT buffer (0.1 M Tris, 10 mM dithiothreitol [DTT]) and incubated for 20 min at 30°C under constant shaking. After reisolation by centrifugation, DTT-treated cells were washed once with Zymolyase buffer (1.2 M sorbitol, 20 mM $K_2HPO_4$, pH 7.4 with HCl), and cell walls were digested for 30 min at 30°C under constant shaking in 7 mL/g pellet Zymolyase buffer containing 1 mg/g pellet Zymolyase 100T. After Zymolyase digestion, cells were reisolated by centrifugation, washed with Zymolyase buffer, and lysed by mechanical disruption in 6.5 mL/g pellet homogenization buffer (0.6 M sorbitol, 10 mM Tris pH 7.4, 1 mM EDTA pH 8.0 with KOH, 0.2% bovine serum albumin, 1 mM phenylmethylsulfonylfluoride) at 4°C. Cell debris were removed from the homogenate twice by centrifugation at 5000 × g for 5 min at 4°C, and mitochondria were pelleted at 14,000 × g for 15 min at 4°C. The mitochondrial pellet was resuspended in SEM buffer (250 mM sucrose, 1 mM EDTA pH 8.0 with KOH, 10 mM 3-(*N*-morpholino)-propansulfonic acid pH 7.2), reisolated by differential centrifugation as described above, and resuspend in SEM buffer, and mitochondria were shock frozen in liquid nitrogen and stored at −80°C.

## In vitro import of radiolabeled proteins

$^{35}$S-radiolabeled Su9-DHFR (*Pfanner et al., 1987*) was synthesized in rabbit reticulocyte lysate (TNT SP6 Quick Coupled Transcription/Translation System, Promega) according to the manufacturer's description. For in vitro protein import reactions, mitochondria were diluted in import buffer (3% BSA, 250 mM sucrose, 80 mM KCl, 5 mM MgCl$_2$, 5 mM methionine, 10 mM KH$_2$PO$_4$, 10 mM MOPS, pH 7.2 with KOH) supplemented with 5 mM creatine phosphate, 0.1 mg/mL creatine kinase, 2 mM ATP, and 2 mM NADH. Eight micromolar antimycin A, 1 µM valinomycin, and 20 µM oligomycin (AVO) were used to dissipate the membrane potential in negative controls. Samples were pre-warmed to 25°C, and the import was started with 3% (v/v) radiolabeled precursor and stopped by transfer on ice and addition of AVO after the indicated times. Subsequently, non-imported prepro-teins were removed by treating samples with 50 µg/mL proteinase K for 10 min on ice. Protein deg-radation was stopped by addition of 1 mM PMSF, mitochondria were reisolated by centrifugation at 20,000 × g for 10 min at 4°C, and pellets were washed with SEM buffer containing 1 mM PMSF. Sub-sequently, mitochondria were resuspended in Laemmli buffer (125 mM Tris/HCl pH 6.8, 4% SDS, 20% glycerol, 0.5 mg/mL bromophenol blue) containing 1% β-mercaptoethanol and 1 mM PMSF, and proteins were denatured for 5 min at 95°C and subjected to SDS–PAGE. The gel was stained with Coomassie solution (40% methanol, 7% acetic acid, 1% Coomassie brilliant blue R 250), de-stained (20% methanol, 10% acetic acid) to confirm equal protein loading, and dried. Import of radiolabel proteins was analyzed by autoradiography.

## Quantification and statistical analysis

Experiments were repeated at least three times, and all attempts at replication were successful. For all quantifications, number of cells scored is included in the figure legends. Differences in means were compared using two-tailed t-tests at the 5% significance level. No randomization or blinding was used in experiments. All analysis was done with GraphPad Prism version 8.01.

## Acknowledgements

We thank members of the ALH laboratory and Dr. Janet Shaw (Utah) for discussion and manuscript comments, Tom Tedeschi (Utah) for technical assistance, Dr. Nikolaus Pfanner for Tom20, Tim54, Tim50, and Tom70 antisera, Dr. Toshiya Endo for Tom40 antisera, Dr. Carla Koehler for OM45 anti-sera, and Dr. Agnieszka Chacinska for Ilv2 antisera. Funding: Research was supported by NIH grants AG043095 and GM119694 (ALH), and AHA 18PRE33960427 (MHS). ALH was further supported by an American Federation for Aging Research Junior Research Grant, United Mitochondrial Disease Foundation Early Career Research Grant, Searle Scholars Award, and Glenn Foundation for Medical Research Award.

## Additional information

### Funding

| Funder | Grant reference number | Author |
|---|---|---|
| National Institute on Aging | AG043095 | Adam L Hughes |
| National Institute of General Medical Sciences | GM119694 | Adam L Hughes |
| American Federation for Aging Research | | Adam L Hughes |
| United Mitochondrial Disease Foundation | | Adam L Hughes |
| Kinship Foundation | | Adam L Hughes |
| Glenn Foundation for Medical Research | | Adam L Hughes |
| American Heart Association | | Max H Schuler |

The funders had no role in study design, data collection and interpretation, or the decision to submit the work for publication.

## Author contributions
Viplendra PS Shakya, Conceptualization, Data curation, Formal analysis, Investigation, Methodology, Writing - original draft, Writing - review and editing; William A Barbeau, Christina S Knutson, Max H Schuler, Data curation, Formal analysis, Investigation, Methodology; Tianyao Xiao, Data curation, Formal analysis, Investigation, Methodology, Writing - review and editing; Adam L Hughes, Conceptualization, Supervision, Funding acquisition, Project administration, Writing - review and editing

## Author ORCIDs
Viplendra PS Shakya 
William A Barbeau 
Tianyao Xiao 
Adam L Hughes 

## Decision letter and Author response
Decision letter https://doi.org/10.7554/eLife.61230.sa1
Author response https://doi.org/10.7554/eLife.61230.sa2

# Additional files

## Supplementary files
- Supplementary file 1. List of mitochondrial protein fates upon FCCP treatment.
- Supplementary file 2. List of yeast strains used in this study.
- Supplementary file 3. List of oligos used in this study.
- Supplementary file 4. List of plasmids generated in this study.
- Transparent reporting form

## Data availability
All data generated or analyzed during this study are included in the manuscript and supporting files.

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
