## [Decision Letter]

**Acceptance summary:**

Normal activity of Mitochondria requires the import of hundreds of proteins from the cytosol. When import is perturbed and mitochondrial precursor proteins accumulate outside mitochondria, cells suffer from severe proteotoxic stress and respond by activating adaptive programs. However, the precise molecular events of this precursor-induced toxicity are still unknown. In this manuscript, Shakya and colleagues reveal the fate of many mitochondrial precursor proteins when their import into mitochondria is blocked by dissipating the inner membrane potential. Using high-throughput microscopy screening, they show that different mitochondrial proteins are now localized to different cellular compartments (including cytosol, endoplasmic reticulum and nucleus) or are basically depleted from the cell when they cannot be imported into mitochondria. They demonstrate that nuclear-localized proteins are subject to proteasomal degradation and that the mitochondrial targeting signal is crucial for its nuclear quality control.

Overall, this is an exciting study which fills an important gap in our knowledge about the quality control of mitochondrial proteins. It will at the same time serve as a valuable resource for many researchers working in the field. The manuscript is very well suited for a widely-read journal like *eLife*.

**Decision letter after peer review:**

Thank you for submitting your article "The nucleus is a quality control center for non-imported mitochondrial proteins" for consideration by *eLife*. Your article has been reviewed by three peer reviewers, one of whom is a member of our Board of Reviewing Editors, and the evaluation has been overseen by David Ron as the Senior Editor. The following individual involved in review of your submission has agreed to reveal their identity: Johannes M. Herrmann (Reviewer #2).

The reviewers have discussed the reviews with one another and the Reviewing Editor has drafted this decision to help you prepare a revised submission.

Summary:

Mitochondrial biogenesis requires the import of hundreds of proteins from the cytosol. When import is perturbed and mitochondrial precursor proteins accumulate outside mitochondria, cells suffer from proteotoxic stress and respond by activating adaptive programs. However, the precise molecular events of this precursor-induced toxicity are still unknown. In this manuscript, Shakya and colleagues reveal the fate of many mitochondrial precursor proteins when their import into mitochondria is blocked by dissipating the inner membrane potential. Using high-throughput microscopy screening, they show that different mitochondrial proteins are now mislocalized to different cellular compartments (including cytosol, endoplasmic reticulum and nucleus) or are basically depleted from the cell when they cannot be imported into mitochondria. They demonstrate that nuclear-localized proteins are subject to proteasomal degradation and that the mitochondrial targeting signal is crucial for its nuclear quality control.

Overall, this is an exciting study which fills an important gap in our knowledge about the quality control of mitochondrial proteins. It will at the same time serve as a valuable resource for many researchers working in the field. The manuscript is very well suited for a widely-read journal like *eLife*.

Essential revisions:

1) A major weakness of the manuscript is the exclusive reliance on tagged proteins. As a requirement for resubmission we ask that at least in selected key experiments you will demonstrate your findings also for an untagged protein for which antibodies exist (many antibodies against mitochondrial proteins in yeast exist). This is especially important since the GFP tag is large and well-folded and known to slow down translocation – for example, the GFP-tag slows down the import of Ilv2 as can be seen by more precursor accumulating (even in the absence of CCCP) relative to the smaller tag HA. Similar difference between GFP-tagged and HA-tagged forms are observed also for Acp1.

2) The authors use an overexpression of Ilv2-/Cox15-/Lat1-GFP variants to demonstrate the relevance of their MTS for the toxicity, nuclear distribution and QC of these proteins. However, the overexpression of the MTS-containing constructs obviously constitutes a severe stress to the cells already without CCCP treatment (Figure 4G and I). It's likely that these constructs act as "cloggers" of the mitochondrial translocases, similar to what was shown in the Weidberg and Amon, 2018 and Boos et al., 2019 papers. Therefore, what the authors observe in these strains might at least partly be due to secondary effects. This is crucial for the interpretation of these observations. Therefore, the authors should carefully check whether in these strains, the import of other mitochondrial proteins is impaired (precursor accumulation in Western blot, activation of classical stress responses). The key experiments should also be repeated with expression at endogenous levels and ideally without fusion of a large, stably folded GFP (a short tag such as HA or Myc should be ok).

---

## [Author Response]

Essential revisions:1) A major weakness of the manuscript is the exclusive reliance on tagged proteins. As a requirement for resubmission we ask that at least in selected key experiments you will demonstrate your findings also for an untagged protein for which antibodies exist (many antibodies against mitochondrial proteins in yeast exist). This is especially important since the GFP tag is large and well-folded and known to slow down translocation – for example, the GFP-tag slows down the import of Ilv2 as can be seen by more precursor accumulating (even in the absence of CCCP) relative to the smaller tag HA. Similar difference between GFP-tagged and HA-tagged forms are observed also for Acp1.

We thank the reviewers for this very important point. Following the reviewers’ recommendation, we have confirmed many of our key findings by evaluating untagged, endogenous proteins using available antibodies. In each case, the results recapitulate our previous findings that were based on epitope tagged proteins. First, we now show that native Ilv2 localizes to the nucleus in FCCP-treated cells via indirect immunofluorescence (Figure 1C), similar to its GFP and FLAG tagged counterparts. We also find that the steady-state levels of native Ilv2 behave similarly to Ilv2-GFP and Ilv2-HA in the presence and absence of FCCP, including the fact that native Ilv2 precursor levels accumulate upon treatment of cells with FCCP (Figure 1E). Moreover, we show that the precursor form of native Ilv2 has a half-life similar to the GFP- and HA-tagged forms upon FCCP treatment (Figure 3—figure supplement 1F), and that the turnover of native Ilv2 is dependent on San1, Ubr1, and Doa10 (Figure 3—figure supplement 1C). Finally, we show that overexpression of untagged Ilv2 leads to MTS-dependent toxicity in the same manner as its GFP tagged counterpart (Figure 5—figure supplement 1A).

In addition to these new experiments with native Ilv2, we also confirmed the localization and steady-state abundance of several proteins from our screen using antibodies against the endogenous proteins. We confirmed mitochondrial and ER localization for Tom20 and OM45, respectively, via indirect immunofluorescence (Figure 2—figure supplement 1B). We also confirmed the prior observed changes in steady-state abundance upon FCCP treatment with antibodies directed against native Ilv2 (Class 1), Tom70 (Class 2), Tom20 (Class 2), Idh1 (Class 3), OM45 (Class 4), Tim54 (Class 5), Tim50 (Class 5), and Tim18 (Class 5) (Figure 2—figure supplement 2B). Steady-state abundance of the GFP-tagged versions of these proteins +/- FCCP has been provided in Figure 2—figure supplement 2A for comparison. Overall, while we certainly cannot rule out that the GFP tag may interfere with the localization and behavior of some of the mitochondrial proteins in our original screen, it does appear that the major conclusions drawn previously about the role of the nucleus in Ilv2 degradation and many fates for non-imported proteins are supported by our new observations. We added additional text in the Results to indicate that not all proteins from the screen have been validated with endogenous antibodies.

2) The authors use an overexpression of Ilv2-/Cox15-/Lat1-GFP variants to demonstrate the relevance of their MTS for the toxicity, nuclear distribution and QC of these proteins. However, the overexpression of the MTS-containing constructs obviously constitutes a severe stress to the cells already without CCCP treatment (Figure 4G and I). It's likely that these constructs act as "cloggers" of the mitochondrial translocases, similar to what was shown in the Weidberg and Amon, 2018 and Boos et al., 2019 papers. Therefore, what the authors observe in these strains might at least partly be due to secondary effects. This is crucial for the interpretation of these observations. Therefore, the authors should carefully check whether in these strains, the import of other mitochondrial proteins is impaired (precursor accumulation in Western blot, activation of classical stress responses). The key experiments should also be repeated with expression at endogenous levels and ideally without fusion of a large, stably folded GFP (a short tag such as HA or Myc should be ok).

We completely agree with the reviewers’ excellent point that our overexpressed MTS-containing mitochondrial proteins may act as cloggers, and that this may contribute to their toxicity. To test this hypothesis, we conducted a series of additional experiments evaluating the impact of Ilv2-GFP overexpression (and an MTS-less variant) on mitochondrial import both in vivo and in vitro. We used both Ilv2-GFP and untagged GFP in different versions of these assays, as we found that both cause similar levels of toxicity when overexpressed (see new Figure 5—figure supplement 1A). As can be seen in new Figure 6A, we found that overexpression of full-length Ilv2-GFP modestly blunted the in vitro import of the classically utilized Su9-DHFR substrate into isolated mitochondria. Deletion of Ilv2’s MTS prevented this effect. Moreover, Ilv2-GFP overexpression led to a small reduction in the steady-state abundance of Tim54 in purified mitochondria, but not matrix protein Aco1 (Figure 6B). Both of these effects were not as strong as the impact of treating cells with FCCP, which severely reduced Su9-DHFR import and Tim54 and Aco1 steady-state abundance in isolated mitochondria (Figures 6C-D). Consistent with a modest import defect upon Ilv2 overexpression, we observed a small to moderate increase in the amount of Idh1 that now localized in the cytoplasm in cells overexpressing Ilv2 (Figures 6E-F, ~10% in WT cells, and ~30% in Ilv2-OE cells). For comparison, Idh1 mis-localized to the cytoplasm in nearly 100% of FCCP-treated cells, in agreement with our original screen data (Figure 6E-F). Overall, these data suggest that Ilv2 OE leads to a modest block in mitochondrial import, but not as robust as in cells treated with FCCP. Given the severe growth defect observed in Ilv2 OE strains compared to growth of cells on FCCP, it seems likely that some of the toxicity associated with Ilv2 OE is independent of its impact on mitochondrial import.

To test this further, we constructed a strain expressing a form of Ilv2 that cannot localize the nucleus. We did this by inserting a nuclear export signal at the C-terminus of Ilv2 on the chromosome, just before an in-frame GFP. As shown in Figures 4F-G, this version of Ilv2, Ilv2-NES, is constitutively excluded from the nucleus, even in the presence of FCCP, and not degraded upon FCCP treatment (Figure 4H). Interestingly, even at endogenous levels, this version of Ilv2 is highly toxic to cells, even more so than Ilv2 OE (Figure 4I). This result suggests that nuclear localization is required to prevent Ilv2 toxicity. Interestingly, we analyzed the impact of Ilv2-NES on mitochondrial import, and found that it had no effect on in vitro import of Su9-DHFR and steady-state the abundance of mitochondrial proteins (Figures 6G-H). Thus, the toxicity of Ilv2 does not appear to entirely correlate with its ability to act as a clogger of mitochondrial import. We have added all of this new data to the manuscript where we indicated above, and have now concluded that nuclear sequestration of Ilv2 is important to mitigate its toxicity in cells, and that toxicity likely results from a combination of mitochondrial import clogging and other effects on the cell that remain unclear at this time.